# Impact of Graft Size Matching on the Early Post-Transplant Complications and Patients Survival in Children after Living Donor Liver Transplantations

**DOI:** 10.3390/children8070579

**Published:** 2021-07-06

**Authors:** Marek Stefanowicz, Grzegorz Kowalewski, Marek Szymczak, Waldemar Patkowski, Krzysztof Zieniewicz, Ireneusz Grzelak, Adam Kowalski, Hor Ismail, Diana Kamińska, Piotr Kaliciński

**Affiliations:** 1Department of Pediatric Surgery and Organ Transplantation, The Children’s Memorial Health Institute Warsaw, 04-730 Warszawa, Poland; m.stefanowicz@ipczd.pl (M.S.); m.szymczak@ipczd.pl (M.S.); a.kowalski@ipczd.pl (A.K.); h.ismail@ipczd.pl (H.I.); p.kalicinski@ipczd.pl (P.K.); 2Department of General Surgery, Transplantation and Liver Surgery, Warsaw Medical University, 02-091 Warsaw, Poland; waldemar.patkowski@gmail.com (W.P.); krzysztof.zieniewicz@wum.edu.pl (K.Z.); chireg@o2.pl (I.G.); 3Department of Gastroenterology, Hepatology, Nutritional Disorders and Pediatrics, The Children’s Memorial Health Institute, 04-730 Warszawa, Poland; diana.kam@wp.pl

**Keywords:** graft to recipient weight ratio, liver transplantation, living donor

## Abstract

We aimed to assess the impact of the graft-recipient weight ratio (GRWR) on early post-transplant complications and patient survival rates in children after living donor liver transplantation (LDLT). We retrospectively analyzed 321 patients who underwent LDLT from 2004 to 2019. The recipients were categorized into four groups: 37 patients had a GRWR ≤ 1.5% (Group A), 196 patients had a GRWR > 1.5% and ≤3.5% (Group B), 73 patients had a GRWR > 3.5% and <5% (Group C) and 15 patients had a GRWR ≥ 5% (Group D). Incidence of early surgical complications including vascular complications, biliary complications, postoperative bleedings, gastrointestinal perforations and graft loss were comparable among groups with a different GRWR. Delayed abdominal wound closure was more common in patients with a GRWR > 3.5%. Recipients with a GRWR < 5% had a significantly better prognosis concerning patients and graft survival. Using grafts with a GRWR < 5% allows us to expand the donor pool and decrease the risk of mortality while on the waiting list, when patients at the time of transplantation have less advanced liver disease. LDLT with a GRWR ≥ 5% is related to a higher risk of poor outcome, and thus should be an option for treating selected patients when the risk of a delayed transplantation is high and access to deceased donors is limited.

## 1. Introduction

There are many technical variants of liver transplantation in children. Selection of the method of liver transplantation depends on organ availability, urgency of surgery, the technical possibilities of transplant centers and their experience with liver surgery and organ transplantation. The shortage of pediatric donors and progress in liver surgery techniques has led to attempts to transplant livers from living donors. Since the first successful living donor liver transplantation (LDLT) in 1989 (completed by Russell Strong in Australia), the method has become an accepted treatment for children with end-stage liver failure, liver tumors and metabolic disorders [1]. 

A size match disparity between a liver graft and the recipient can result in complications after LDLT. The graft-to-recipient weight ratio (GRWR) is a commonly used index to predict size mismatching. Patients with too small grafts may develop encephalopathy, coagulopathy, cholestasis or acute renal failure with poor graft survival [2]. The small for size graft situation is more common in adults undergoing LDLT and is defined by a GRWR < 1% or even less [3,4]. Anatomical and physiological characteristics of the liver are different in children and adults. The liver volume to body weight ratio in children decreases with age until approximately 16 years [5]. Due to this physiological phenomenon, the criteria of small for size graft for pediatric patients are different from those for adult patients, especially in infants and young children. As opposed to adults, in pediatric LDLT settings, liver grafts are more often too large for the recipient, resulting in the “large for size” syndrome with poor graft perfusion, increased risk of vascular thrombosis and graft dysfunction as well as difficulty with abdominal wound closure, among other complications [6,7,8]. 

The optimal and acceptable range of the GRWR for pediatric LDLT should be determined. The aim of our study was to assess the impact of the GRWR on the early post-transplant complications, patients and graft survival in children after LDLT.

## 2. Materials and Methods

We performed a retrospective analysis of 321 children after primary living donor liver transplantation was completed in our department between 1 January 2004 and 31 December 2019. Patients after LDLT retransplantation were excluded from the study. Donor and recipient data were collected, including demographic and clinical details from their hospital records and data from the National Transplant Registry. Graft characteristics and the GRWR index were obtained for each patient from operative protocols. To assess the impact of graft size on recipient outcomes, patients were categorized into four Groups: patients with a GRWR≤ 1.5% (Group A), patients with a GRWR > 1.5% and ≤3.5% (Group B), patients with a GRWR > 3.5% and <5% (Group C) and patients with a GRWR ≥ 5% (Group D). The cut-point analysis is described in the statistical analysis section.

Variables related to the recipient, donor and surgery were analyzed. Early postoperative surgical complications (<30 days), were evaluated, including hepatic artery thrombosis (HAT), portal vein thrombosis (PVT), biliary leakage from biliary anastomosis or cut surface of the liver, postoperative bleeding and gastrointestinal perforation. The number of patients with a minimum of one episode of acute rejections (AR) during all follow ups was assessed. Time in postoperative ICU after transplantation, hospital stay, the need for delayed abdominal wound closure and time of closure, early and late graft and patient survival were assessed and compared between study groups. 

### 2.1. Donors

Grafts were procured from healthy adult donors who provided informed consent. A fragment of the liver was procured from a living donor in an adult transplantation surgery unit with extensive experience in liver surgery which cooperates with our center. Computer tomography with three-dimensional reconstruction, ultrasound examination and magnetic resonance cholangiography were performed to evaluate hepatic vascular and biliary anatomy. In all donors, biliary anatomy was also evaluated using intraoperative real-time cholangography. Currently, to estimate liver volumes, we use Vitrea software (Vital^®^ Images, Inc., Minnetonka, MN, USA), which allows for perioperative prediction of graft weight and GRWR. 

### 2.2. Recipient Surgery

The liver graft was implanted into the recipient’s abdominal cavity using a piggyback technique. The anastomosis of the left hepatic vein was sutured in a triangular fashion to confluence the left and middle hepatic vein, with three running non-absorbable 6/0 monofilament sutures. The portal vein anastomosis was sutured continuously in an end-to-end fashion with a 6/0 or 7/0 absorbable suture. The hepatic artery of the graft was anastomosed end-to-end to the hepatic artery of the recipient with non-absorbable 8/0 interrupted sutures. During vascular anastomosis, the graft was flushed via a portal vein with a 2.5% cold albumin solution. Blood flow through the graft was restored after hepatic artery anastomosis. For biliary tract reconstruction, the Roux-en-Y hepaticojejunostomy or duct to duct anastomosis was performed. Intraoperative color Doppler ultrasonography was performed to assess intrahepatic blood flow. 

At the end of operation, to avoid compression of the graft, the type of abdominal closure was determined by the surgeon based on the tension required to reapproximate skin and/or fascia and Doppler ultrasonography. We used Vicryl mesh, which was sutured to fascia and/or skin. After a few days, wound closure was attempted. Hernia repair was performed electively after a minimum of 12 months from liver transplantation.

To prevent vascular thrombosis, routine anticoagulation was used postoperatively in all patients with enoxiparine, followed by acetylsalicylic acid at a dose of 1 mg/kg once a day for 6 months. 

### 2.3. Immunosuppressive Regimens

Before graft reperfusion, methylprednisolone was administered intravenously at a dose of 10 mg/kg. Standard immunosuppression consisted of tacrolimus and mycophenolate mofetil (MMF). Steroids were given only based on individual indications, with individual modifications depending on the patient’s characteristics. The target tacrolimus level was 8–12 ng/mL during the first month. Patients with ABO-incompatible liver transplantation were given triple drug therapy with Il-2 antagonist as an induction in children over 2 years of age.

AR was diagnosed clinically by elevated ASPAT (aspartate aminotransferase) and ALAT (alanine aminotransferase), GGTP (gamma glutamyl transpeptidase) activity, bilirubine concentration and/or low tacrolimus concentration and liver biopsy. In selected patients, empirical treatment was started without biopsy. Typically, metylprednisolon 3–6 boluses were used in the initial treatment of AR.

### 2.4. Statistical Analysis

Data was analyzed using Statistica 13 software developed by StatSoft inc. To determine the best cut-point for improving survival, we evaluated the ability of prognostic stratification at each 0.5 GRWR value using the magnitude of the log-rank test χ^2^ statistic. The cut-point that appeared to provide a statistically significant survival difference between the resulting subgroups and maintain clinical usefulness in terms of subgroups population was taken into consideration. Survival curves were constructed according to the Kaplan–Meier method, and a log-rank or Cox–Mantel test was used to determine whether significant differences were present among survival curves. We used a multivariate analysis to evaluate the validity of GRWR influence on recipients’ survival. Prognostic factors with either known or suspected clinical importance and these with *p* < 0.1 identified in the univariate analysis were incorporated into the multivariate analysis model. Additional statistical analysis involved assessing baseline demographics and clinical data using medians, ranges and distributions for categorical variables. The Student t test and Mann–Whitney U test were used to assess unpaired associations between continuous variables. For a comparison of more than two Groups, the one-way analysis of variance (ANOVA) was used. We compared categorical variables using the Chi-Square Test of Independence. A *p* value of less than 0.05 was considered as statistically significant. 

This study was approved by the Institutional Ethical Committee (approval number: 27/KBE/2021).

## 3. Results

### 3.1. Recipients, Donors and Grafts

Between January 2004 and December 2019, we performed 327 LDLT in 323 patients younger than 18 years of age at our institution. A total of 321 patients received primary LDLT. Follow-up time ranged from 3 days to 17.4 years, median 7.2 years; in 240 patients (74.8%), the follow-up time was longer than 5 years and in 139 patients (43.3%) it was longer than 10 years.

As shown in Table 1, we examined the ability of each GRWR cutoff value to affect survival and found that a GRWR value of 3.5 and 5 most significantly stratified the prognoses of liver transplant recipients. We subsequently divided the analyzed population into 4 groups to further analyze the cohort according to our clinical experience and published medical data. 

Group A consisted of 37 patients with a GRWR ≤ 1.5%; Group B included 196 patients with a GRWR > 1.5% and ≤3.5%; Group C included 73 patients with a GRWR > 3.5% and <5% and 15 patients with a GRWR ≥ 5% were in Group D. 

The underlying liver diseases are listed in Table 2. The most common indication for LDLT in all groups was cholestatic disease 203 pts (63.2%), including 184 patients (57.3%) with biliary atresia after Kasai portoenterostomy before LDLT. Primary liver tumor and acute liver failure were the second and third most common indications for LDLT, in 44 (13.7%) and 27 (8.4%) patients of our study cohort, respectively. Liver tumors included hepatoblastoma in 32 patients, hepatocarcinoma in 6 and other tumors in 6.

The baseline characteristics of the four groups of recipients, donors and grafts are presented in Table 3. Patients from Group C and D were significantly younger than those from Group A and B. All patients in Group D and 83.6% patients in Group C were under 1 year of age. Recipients’ body mass was also significantly lower in Groups C and D. Recipient’s disease severity as measured by the PELD score was significantly higher in Groups C and D. The percentage of patients receiving grafts from AB0-incompatible donors was similar between groups (*p* = 0.226). Urgent liver transplantation was performed significantly less often in patients from Group C than from Groups A, B and D (*p* < 0.003, *p* < 0.006 and *p* < 0.002, respectively). 

Despite the donor age and body mass being similar among the recipient study Groups, there were significant differences in graft weight and GRWR between Groups (*p* < 0.0001 and *p* < 0.05), with Group D characterized by the largest graft weight.

Graft and operative details are listed in Table 4. The grafts consisted of monosegment in 10 patients, left lateral segments (segment II + III) in 277 patients, the left lobe (segments II+III+IV) in 32 patients and the right lobe (V+VI+VII+VIII) in two patients (Figure 1). For biliary tract reconstruction, Roux-en-Y hepaticojejunostomy was performed in 276 patients (86%) and duct to duct anastomosis in 45 cases (14%). Cold ischemia time was different among groups, with Group B characterized by the shortest cold ischemia time.

### 3.2. Postoperative Complications

Postoperative complications in our patients were typical for pediatric LDLT (Table 5). There were 16 patients (5%) who developed hepatic artery thrombosis (HAT). There was, however, no case of HAT in patients from Group D. Although there was a numerical trend toward a greater number of HAT in Group B, it did not reach statistical significance. The incidence of HAT was not different between Groups A, B and C (*p* = 0.338). All 16 patients with early HAT underwent emergency reoperation with a successful restoration of hepatic flow in 13 patients and at the time of discharge, hepatic artery was patent in all 13 patients. In late observation, 12 of the patients with early HAT thrombosis presented good arterial flow. Two patients from Group B developed intrahepatic abscesses after HAT and underwent retransplantation 2 and 5 months after LDLT. Two patients died due to HAT: one after retransplantation from multiple organ dysfunction syndrome (MODS) and the second six days after LDLT following complications related to HAT.

In 27 patients (8.4%), early portal vein thrombosis developed in the postoperative course, but incidence of PVT was comparable among the groups (*p* = 0.229). PVT was detected by Doppler ultrasonography and all these patients underwent emergency thrombectomy, with reanastomosis done in two patients. In 26 patients, early patency of portal vein was reestablished, and portal flow was present in 26 patients at the time of discharge. The patient with PVT from Group A needed retransplantation 10 days after the first LDLT. Five patients with early PVT were diagnosed with late PVT. Four of five patients with late PVT developed complications of portal hypertension and a, porto-systemic shunt was performed on two patients, while one patient underwent splenectomy and one patient underwent spleen embolization. In late observation, portal vein was patent in 21 of patients with early PVT and one patient lived after a second transplantation.

The overall biliary leakage rate of the entire series was 44 (13.7%), with 33 patients (10.3%) developing a biliary fistula from an anastomotic site and 11 patients (3.4%) from a hepatic cut surface. The incidence of biliary leak from the surface of the liver was not different among the groups (*p* = 0.167). There was no biliary leak from anastomosis in patients from Group D. No difference was found in the rate of anastomotic biliary leak among Groups A, B and C (*p* = 0.502). Two patients from Group B, one from Group C and one from Group D underwent liver retransplantation due to biliary complications: 7 years, 11 years, 17 months and 10 months after first LDLT, respectively. One patient from Group C died 8.5 years later from autoimmune thrombocytopenia after second liver retransplantation. 

Postoperative bleeding resulted in relaparotomy in 55 patients (17.1%). The incidence of reoperations due to bleeding was not different between groups (*p* = 0.705). Gastrointestinal perforations were observed only in six patients in Group B and five in Group C, with similar incidence between both groups (*p* = 0.163).

In 130 patients (43.6%), the abdominal wound was closed at the time of LDLT. Delayed abdominal wound closure was similar in groups with GRWR < 3.5% (Group A vs. Group B *p* = 0.249) and groups with GRWR ≥ 3.5% (Group C vs. Group D *p* = 0.264). Delayed abdominal wound closure occurred more frequently in Groups with GRWR ≥ 3.5% and was significantly higher (Group A vs. Group C *p* < 0.01, Group A vs. Group D *p* < 0.01, Group B vs. Group C *p* < 0.02 and Group B vs. Group D *p* < 0.03) (Table 5).

There were differences between the groups in the length of an intensive care unit stay and hospital stay, and the difference proved to be statistically significant (*p* < 0.001 and *p* < 0.01). The ICU stay was longest in patients with the highest GRWR, while hospital stay was longest in patients with the lowest GRWR.

In our cohort, a minimum of one episode of acute cellular or antibody mediated rejection was encountered in 127 patients (39.6%) during follow up and incidence was similar among recipients with different GRWR.

Retransplantation was performed in 15 patients (4.7%) from our cohort during the entire follow-up (Table 5). The incidence of retransplantations was not different among the groups (*p* = 0.364). The most common causes of graft loss were biliary complications in four patients, followed by chronic rejection in three. In most patients (12, 80%) late retransplantation, more than 30 days after primary LDLT, was done. In three patients, early retransplantation was done due to hyperacute rejection in two patients and PVT in one (Table 6).

### 3.3. Patient and Graft Survival

In total 36/321 patients died (mortality rate: 11.2%). Patient survival after 1, 5 and 10 years was: in Group A 92%, 88% and 85%, in Group B 93%, 92% and 91%, in Group C 95%, 90% and 85%, in Group D 73%, 67% and 55%, respectively. In Group A graft survival was 86%, 85% and 85%, in Group B 91%, 90% and 85%, in Group C 95%, 89% and 85%, in Group D 67%, 60% and 45% at 1, 5 and 10 years, respectively. Kaplan–Meier Curves depicting patients and graft survival, as well as calculated survival estimates with *p*-values, are shown in Figure 2 and Figure 3. No statistically significant difference in patient and graft survival was found between Groups A, B and C. Patient and graft survivals were significantly worse in Group D (*p* < 0.05).

A logistic regression model was created to predict the risk of mortality. Analysis suggests that the GRWR is an independent risk factor of death in children after LDLT (OR = 1.39; 95%CI 1.02–1.89) with a cut-off value of 5 (OR = 5.806; 95%CI 1.77–19-07). The multivariate analysis model is shown in Table 7. 

The causes and time of patient deaths in each group are shown in Table 8. Of the 36 patients (11.2%) who died, the most common cause was infection in 11 cases (26.8%), followed by multiple organ dysfunction syndrome (MODS) in 9 (25%). A total of 15 patients died in the early postoperative period (<30 days). Four patients died after liver retransplantation, but only one within 30 days after the second transplantation. Of the 10 patients with reduced grafts, one died from Group C and one from Group D, with the highest GRWR in our cohort (10.68) due to infectious complications and MODS, respectively.

## 4. Discussion

LDLT in the pediatric population is an effective therapeutic option, as the availability of grafts from deceased donors for small children is very limited. It reduces or eliminates waitlist death, shortens time to transplantation and provides better outcomes for children [9]. Obtaining well size-matched liver grafts for many younger patients, however, especially for those under 1 year of age, still remains a problem.

The GRWR cutoff value varies among studies. Kiuchi et al. reported 276 patients after LDLT that were categorized into five groups: extra-small for size (GRWR < 0.8%), small (0.8% ≤ GRWR < 1%), medium (1% ≤ GRWR < 3%), large (3% ≤ GRWR < 5%) and extra-large (GRWR ≥ 5%) [3]. In their study, pediatric and adult patients were analyzed together. They concluded that using grafts with a GRWR < 1% leads to lower graft survival due to enhanced parenchymal cell injury and reduced metabolic and synthetic capacity, reflected by hyperbilirubinemia and coagulopathy. A negative impact of LDLT with a GRWR ≥ 5% was less pronounced. In a study by Li et al., 252 patients underwent LDLT and were categorized into three Groups by GRWR; the authors concluded that the GRWR in pediatric LDLT is a major risk factor that affects survival and recommended a GRWR between 2% and 4% as the optimal range [8]. Goldaracena et al. reported that pediatric LDLT with grafts having a GRWR ≥ 2.5%, and even >4%, can be performed safely with similar results as graft having a GRWR < 2.5% [10]. They also observed a higher rate of delayed abdominal wall closure that did not impact on the overall outcome. In another study, the authors concluded that a GRWR between 1.9% and 5.8% would not cause noticeable adverse events for infantile LDLT recipients ≤8 kg [11].

In the current study, we demonstrated that LDLT can be performed safely in children with grafts having a GRWR < 5%. In patients with GRWR < 5%, we achieved excellent patients and graft long-term outcomes 1, 5 and 10 years after LDLT, compared to other studies [9]. In contrast, patients with a GRWR ≥ 5% had significantly worse graft and patient survival in the same period of time. Bonatti et al. reported that in patients with a too large graft, inadequate perfusion of graft may result in graft dysfunction [12]. Large for size grafts in small children may increase the risk of vascular complications [13]. A higher GRWR is an independent risk factor of HAT [14]. In another study, hepatic artery stenosis/thrombosis was more frequently observed in patients with GRWR ≥ 4% [8]. As reported by Moon, GRWR ≥ 4% and a portal vein size <5 mm is a risk factor for portal vein complications [15]. We did not evaluate the size of portal veins, but noted that the majority of patients, especially with a GRWR > 3.5% had biliary atresia, and biliary atresia is associated with portal vein hypoplasia and sclerosis, which may cause difficulties during portal vein anastomosis and predispose patients towards thrombosis [16]. 

In our study, we have not observed significant differences in the incidence of early vascular complications including HAT and PVT among groups with a different GRWR. The most severe complications related to HAT occurred in patients with size matched grafts and a GRWR >1.5% and ≤3.5%. Incidence of HAT and PVT was similar to thaT described by other authors [17,18]. Our strategy to resolve early vascular complications was immediate surgical treatment. All patients with HAT or PVT underwent emergency thrombectomy and when necessary reanastomosis. We were able to restore blood flow in the VAST majority of patients. 

There are many factors contributing to biliary complications, including anatomical variations, inadequate arterial supply, ischemia reperfusion injury and immunological reactions. Biliary leaks occur in 5.1–23.4% of patients [19]. Li et al. showed that intestinal fistula and bile leakage was more frequently observed in patients with GRWR < 2% [20]. In our material, incidence of biliary leakage from an anastomotic site or liver cut surface was similar to that described in the literature and was not different between study groups.

In our patients, incidence of early postoperative bleeding was 17% and was higher than described in other studies. Okada et al. reported that 3.4% patients after pediatric LDLT underwent early relaparotomy due to postoperative bleeding and in a study by Hara et al., 10.2% patients after adult to adult LDLT needed laparotomy due to postoperative bleeding [21]. In those studies, an indication for relaparotomy was bleeding with hemodynamic instability. In our study, additional indications for surgical intervention were an increased number of transfusions in a short period of time and removal of hematoma to prevent infection complications. 

As reported in the literature, 9–29% of pediatric liver transplant recipients require retransplantation [22,23]. According to the European Liver Transplant Registry which has collected data concerning 146,782 LTs in 132,466 patients from 169 centers and 32 countries, about 10% of patients needed retransplantation [20]. In our cohort, only 4.7% of patients underwent second liver transplantation. Incidence of graft loss was similar in all study groups. Also, indications and the number of patients who needed early or late retransplantation were similar to those described in other studies. 

As expected, patients with a higher GRWR > 3.5% were significantly younger and therefore had a significantly lower body mass at the time of LDLT. Graft weight was the highest in patients with a GRWR ≥ 5%. In patients with a higher GRWR, graft compression and compartment syndrome may cause deterioration of vascular flow, vascular thrombosis or graft dysfunction [7]. Delayed abdominal wound closure was our routine strategy to prevent these complications. In our cohort, a higher rate of delayed abdominal wound closure was observed in patients with a higher GRWR > 3.5%. Comparably, other authors reported a 45% incidence of delayed abdominal wound closure in patients with a GRWR ≥ 4% [10]. 

Another strategy that allowed us to overcome the problem of too large graft was an additional reduction of left lateral segments (LLS) to decrease GRBR and to reshape a too thick graft. Kitajima et al. have shown in their study that reduction of grafts was considered when the estimated GRWR exceeded 4% [7]. Depending on the size and shape of a graft and ratio of maximum thickness of graft to the anterior posterior diameter of a recipient’s cavity, they used non anatomically reduced (ratio < 1.0) and reduced thickness left lateral segments (ratio ≥ 1.0). They observed that portal vein flow was increased and delayed abdominal wall closure was needed less in patients with a reduced thickness graft compared to non-anatomically reduced graft (2% vs. 28%). They showed a mortality or graft loss in 18% of cases. In other series of patients transplanted with reduced left lateral segments or monosegmental grafts, up to 40% of patients still required delayed abdominal wall closure [24]. We used monosegmental graft in a limited number of patients with a similar mortality rate.

In our cohort, 40% of patients after LDLT had a minimum of one episode of acute rejection (cellular and antibody mediated) during all follow-up periods. These results are similar to other studies. Kehar et al. reported that the 1, 3 and 5 year acute cellular rejection free survival rates in the group of 135 patients after LDLT were 64.4%, 61.1% and 61.1%, respectively [9].

In adult LDLT, small for size syndrome is a well-known complication. The main factor is portal vein hypertension causing graft hyper perfusion. Portal vein inflow modulation by splenectomy, splenic artery ligation or porto-systemic shunts are surgical strategies to prevent small for size syndrome [25]. In our study, small for size syndrome is a minor problem. Only three patients had a GRWR < 1%. Portal vein inflow modulation was not routinely considered in our patients. The most common indication for LDLT in this group was biliary atresia, which coexists with portal vein hypoplasia. In these patients, one of the major problems during LDLT is adequate portal inflow. In our institution, splenectomy is a procedure reserved for patients with splenomegaly and a low white blood cell count (<2000–3000/mm^3^) and platelet count (<50,000/mm^3^). Additionally, splenectomy increases the risk of PVT after LDLT [26]. In our cohort one patient with a GRWR ≤ 1.5% had banding of their portal vein performed as an emergency procedure due to graft congestion after reperfusion (hepatic vein stenosis was excluded based on Doppler ultrasonography). 

Our study indicated that a GRBR over 5% poses a significantly increased risk of graft loss and to a recipient’s mortality, although this is not directly related to typical surgical complications. The multivariate regression analysis model created in our research included surgical risk factors such as HAT, PVT, biliary leak and postoperative bleeding—nevertheless, the GRWR remained an independent risk factor of death in patients after LDLT. The outcome is most likely the multifactorial effect of graft hypoperfusion and dysfunction, prolonged ventilatory support and ICU stays, secondary infections, etc. The most common cause of death in this group was multiple organ dysfunction syndrome followed by infectious complications. 

This study has several limitations. The main limitation is its retrospective nature. The second limitation is the relatively small numbers of patients with an extremely low or high GRWR in our cohort. We did not assess recipients’ abdominal volumes and differences in shape of liver grafts, especially anteroposterior thickness of graft, which is particularly problematic in smaller children without hepatomegaly or ascites. This did not allow us to assess how these variables could affect the results of using large grafts. We also did not analyze late complications (hepatic artery, portal vein or biliary stenosis) and their impact on outcome in this paper. We also were not able to determine a safe lower limit of the GRWR in pediatric patients. 

In conclusion, recipients of LDLT with a GRWR < 5% had significantly better prognosis concerning patients and graft survival. Using grafts with a GRWR < 5% allows us to expand the donor pool and decrease risk of mortality while on the waiting list, meaning patients at the time of transplantation have less advanced liver disease, thus the incidence of complications in posttransplant period is reduced. LDLT with a GRWR ≥ 5% is related to a higher risk of poor outcomes, and thus should be an option for treating selected patients when the risk of delaying transplantation is very high and access to deceased donors is limited. It should be considered, however, if and which technique of graft reduction should be used in this situation. 

## Figures and Tables

**Figure 1 children-08-00579-f001:**
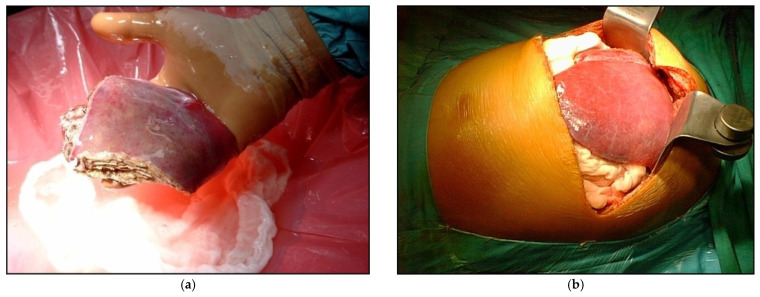
Example of LDLT with different types of graft (**a**) Reduced graft after backtable preparation; (**b**) During the recipient operations, the left lateral segments after reperfusion.

**Figure 2 children-08-00579-f002:**
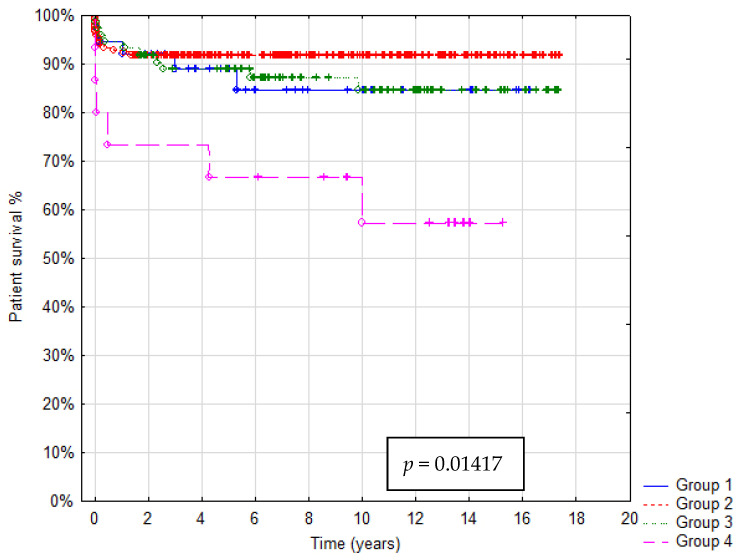
Kaplan–Meier patient survival curve.

**Figure 3 children-08-00579-f003:**
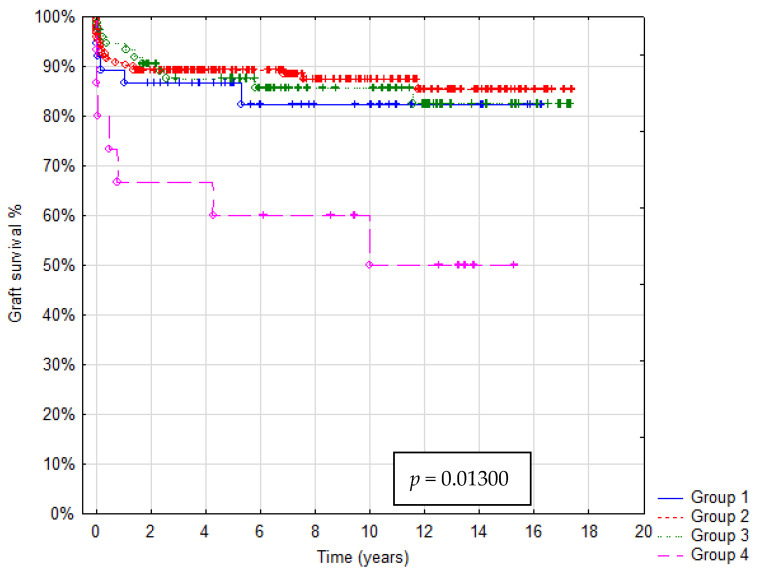
Kaplan–Meier graft survival curve.

**Table 1 children-08-00579-t001:** Cut-point analysis.

GRWR Cutoff Value	Number of Patients	Overall Survival (%)	χ^2^	Log-Rank Test *p*-Value
<1 vs. ≥ 1	3 vs. 317	100 vs. 88.33	0.00013588	0.54246
<1.5 vs. ≥ 1.5	37 vs. 283	86.11 vs. 88.73	0.20301664	0.65271
<2 vs. ≥ 2	93 vs. 227	89.25 vs. 88.11	0.06631477	0.79707
<2.5 vs. ≥ 2.5	143 vs. 177	90.21 vs. 87.01	0.64005114	0.42278
<3 vs. ≥ 3	184 vs. 136	91.30 vs. 84.56	3.03766444	0.07729
<3.5 vs. ≥ 3.5	232 vs. 88	90.95 vs. 81.82	4.18125661 *	0.03652
<4 vs. ≥ 4	271 vs. 49	89.30 vs. 83.67	0.97788367	0.31531
<4.5 vs. ≥ 4.5	293 vs. 27	89.42 vs. 77.78	2.74998956	0.08962
<5 vs. ≥ 5	305 vs. 15	89.84 vs. 60.00	11.0518162 *	0.00055

GRWR graft-recipient weight ratio, * corresponds to the cut points with the highest χ^2^ statistic.

**Table 2 children-08-00579-t002:** Underlying liver disease of patients.

Diagnosis	Group A (*n* = 37) GRWR ≤ 1.5%	Group B (*n* = 196) >1.5% GRWR ≤ 3.5%	Group C (*n* = 73) >3.5% GRWR < 5%	Group D (*n* = 15) GRWR ≥ 5%	Total (*n* = 321)
Cholestatic disease *Biliary atresia*	15 (40.6%)	120 (61.2%)	56 (76.7%)	12 (80%)	203 (63.2%)
	13 (35.1%)	104 (53.1%)	55 (75.3%)	11 (73.3%)	184 (57.3%)
Liver tumors	4 (10.8%)	33 (16.8%)	6 (8.3%)	1 (6.7%)	44 (13.7%)
Acute liver failure	4 (10.8%)	21 (10.7%)	0	2 (13.3%)	27 (8.4%)
Metabolic disorders	2 (5.4%)	6 (3.1%)	2 (2.7%)	0	10 (3.2%)
Others	12 (32.4%)	16 (8.2%)	9 (12.3%)	0	37 (11.5%)

**Table 3 children-08-00579-t003:** Characteristics of donors and recipients according to GRWR.

Characteristics	Group A (*n* = 37) GRWR ≤ 1.5%	Group B (*n* = 196) >1.5% GRWR ≤ 3.5%	Group C (*n* = 73) >3.5% GRWR < 5%	Group D (*n* = 15) GRWR ≥ 5%	*p*-Value
Recipients					
Age (months), median (range) <1 year	78 (5–184)	17 (5–176)	8 (3–29)	6 (0.5–11)	*p* < 0.001
	1 (2.7%)	62 (31.6%)	61 (83.6%)	15 (100%)	
Body mass (kg); median (range)	22.5 (12.8–45)	10 (4–47)	7 (4.5–12.3)	5.6 (3.1–7.8)	*p* < 0.001
PELD score median (range)	7 (−9–38)	13 (−11–41)	21 (−9–38)	20 (4–42)	*p* < 0.0001
AB0i LDLT	4 (10.8%)	29 (14.8%)	16 (21.9%)	4 (26.7%)	0.226
Urgent LDLT	6 (16.2%)	24 (12.2%)	1 (1.4%)	3 (20%)	*p* < 0.021
Donors					
Age (year); median (range)	33 (21–57)	31 (19– 54)	31 (18–56)	32 (21–44)	*p* < 0.05
Body mass (kg); median (range)	67.5 (50–106)	65 (45–110)	68 (46–106)	67 (54–90)	*p* = 0.2849
Graft weight (g); median (range)	280 (158–615)	252 (131–919)	275 (198–506)	320 (212–507)	*p* < 0.0001
GRWR (%); median (range)	1.35 (0.86–1.5)	2.42 (1.51–3.49)	3.98 (3.52–4.94)	5.54 (5–10.68)	*p* < 0.05

**Table 4 children-08-00579-t004:** Surgical variables.

Operation	Group A (*n* = 37) GRWR ≤ 1.5%	Group B (*n* = 196) >1.5% GRWR ≤ 3.5%	Group C (*n* = 73) >3.5% GRWR < 5%	Group D (*n* = 15) GRWR ≥ 5%	Total (*n* = 321)
Graft type					
Monosegment	0	4	5	1	10 (3.1%)
Left lateral segment	17	178	68	14	277 (86.3%)
Left lobe	19	13	0	0	32 (10%)
Right lobe	1	1	0	0	2 (0.6%)
Biliary anastomosis					
Roux-en-Y hepaticojejunostomy	28	164	70	14	276 (86%)
Duct to duct anastomosis	9	32	3	1	45 (14%)
Cold ischemic time (min); median (range)	296 (210–450)	265 (183–485)	270 (200–383)	287 (240–315)	

**Table 5 children-08-00579-t005:** Recipients’ posttransplant outcomes following LDLT according to the GRWR (n,%).

	Group A (37) GRWR ≤ 1.5%	Group B (196) >1.5% GRWR ≤ 3.5%	Group C (73) >3.5% GRWR < 5%	Group D (15) GRWR ≥ 5%	Total	*p*-Value
Hepatic artery thrombosis						
Early (%)	1 (2.7%)	13 (6.6%)	2 (2.7%)	0	16 (5.0%)	*p* = 0.338
Rethrombosis after trombectomy (%)	0	2 (1%)	1 (1.4%)	0	3 (0.9%)	
Portal vein thrombosis						
Early (%)	4 (10.8%)	12 (6.1%)	10 (13.7%)	1 (6.7%)	27 (8.4%)	*p* = 0.229
Rethrombosis after trombectomy (%)	1 (2.7%)	3 (1.5%)	2 (2.7%)	0	5 (1.6%)	
Biliary leaks						
Biliary anastomosis	6 (16.2%)	19 (9.7%)	8 (11%)	0	33 (10.3%)	*p* = 0.502
Cut surface of liver	1 (2.7%)	5 (2.6%)	3 (4.1%)	2 (13.3%)	11 (3.4%)	*p* = 0.167
Postoperative bleeding	6 (16.2%)	34 (17.3%)	14 (19.2%)	1 (6.7%)	55 (17.1%)	*p* = 0.705
Gastrointestinal perforations	0	6 (3.1%)	5 (6.8%)	0	11 (3.4%)	*p* = 0.163
Acute rejections	14 (37.8%)	77 (39.5%)	31 (42.5%)	5 (33.3%)	127 (39.6%)	*p* = 0.907
Abdominal wound closure *						
Primary	20 (58.8%)	86 (48.0%)	22 (30.6)	2 (15.4%)	130 (43.6%)	
Delayed Time to closure (days)	14 (41.2%) 6 (2–9)	93 (52.0%) 5 (2–65)	50 (69.4%) 7 (1–63)	11 (84.6%) 5 (4–123)	168 (56.4%)	*p* < 0.005
ICU stay (days) *; median (range)	3 (2–27)	4(2–62)	5 (2–40)	10 (3–48)		*p* < 0.001
Hospital stay (days) *; median (range)	48.5 (17–324)	35 (13–136)	40 (12–167)	37 (24–189)		*p* < 0.01
Re-Tx (%)	2 (5.4%)	9 (4.6%)	2 (2.7%)	2 (13.3%)	15 (4.7%)	*p* = 0.364

* Patients who underwent retransplantation or died during a hospital stay after LDLT were excluded.

**Table 6 children-08-00579-t006:** Causes and time (days or months) of retransplantation.

Cause of Retransplantation	Group A (*n* = 37) GRWR ≤ 1.5%	Group B (*n* = 196) >1.5% GRWR ≤ 3.5%	Group C (*n* = 73) >3.5% GRWR < 5%	Group D (*n* = 15) GRWR ≥ 5%	Total (*n* = 15)
HAT	0	2 (2.4 months)	0	0	2 (13.3%)
PVT	1 (10 days)	0	0	0	1 (6.7%)
Biliary complications	0	2 (83.131 months)	1 (17 months)	1 (10 months)	4 (26.7%)
Hyperacute rejection	1 (2 days)	1 (3 days)	0	0	2 (13.3%)
Chronic rejection	0	2 (5.16 months)	0	1 (48 months)	3 (20%)
Other causes	0	2 (2.91 months)	1 (139 months)	0	3 (20%)

HAT hepatic artery thrombosis, PVT portal vein thrombosis.

**Table 7 children-08-00579-t007:** Risk factors for recipient death in multivariate analysis.

Variables	Adjusted OR (95% CI)	*p*-Value
Urgent LDLT	1.23 (0.36–4.18)	0.744
Recipient age at LDLT	1.00 (0.97–1.04)	0.860
Recipient body mass at LDLT	1.02 (0.86–1.20)	0.859
GRWR	1.39 (1.02–1.89)	0.036
GRWR ≥ 5	5.81 (1.77–19-07)	0.004
PELD/MELD	1.01 (0.98–1.03)	0.507
Postoperative bleeding	1.74 (0.75–4.03)	0.196
PVT	1.85 (0.63–5.40)	0.264
HAT	1.05 (0.21–5.19)	0.953
Biliary anastomosis leak	0.9 (0.25–3.19)	0.868

PELD pediatric end-stage liver disease, MELD model for end-stage liver disease.

**Table 8 children-08-00579-t008:** Causes and time (days or months) of patient death.

Cause of Death	Group A (37) GRWR ≤ 1.5%	Group B (196) >1.5% GRWR ≤ 3.5%	Group C (73) >3.5% GRWR < 5%	Group D (15) GRWR ≥ 5%
Infections	0	2 (10.15 days)	7 (8,30 days, 3,9,13,28,31,70 months)	2 (22 days, 6 months)
MODS	1 (13 months)	5 (4.11 * days, 2.2 *,8 months)	0	3 (3.4 days, 51 * months)
Malignancy/tumor recurrence	0	4 (30 days, 2,4,17 months)	1 (5 months)	0
Acute rejection	0	1 (30 days)	0	0
GVHD	0	1 (3 months)	0	0
HAT	0	1 (6 days)	0	0
Central nervous system complications	2 (30 days, 36 months)	2 (4.7 days)	0	0
Gastrointestinal bleeding	0	0	0	1 (120 months)
Allograft dysfunction	1 (75 months)	0	0	0
Autoimmune thrombocytopenia	0	0	1 (120 * months)	0
Non-medical	0	0	1 (3 months)	
Total (%)	4 (10.8)	16 (8.2)	10 (13.7)	6 (40.0)

MODS multiple organ dysfunction syndrome, GVHD graft versus host disease. * Patient died after retransplantation.

## Data Availability

Most relevant data are within the paper. Most the output data were taken from the Polish National Transplant Registry at https://rejestrytx.gov.pl/tx/ (accessed date 1 June 2021). Since the data collected in the Registry is sensitive and thus, protected by law (the Act on Personal Data Protection and the Medical Records Act), access to the database is limited; it can be accessed only upon meeting registration criteria. The Registry is under supervision of Polish Transplant Coordinating Centre “Poltransplant”, a budget funded unit of the Polish Ministry of Health. Since the registry is only available to a limited number of healthcare professionals working in transplantation units across Poland, access to the database is impossible for people not involved directly in transplantation and coordination processes on the national level. All of the patients’ medical histories and other vital information used in the creation of the database are available directly at the Children’s Memorial Health Institute, Warsaw, Poland after contacting the Director of Scientific Affairs, MD PhD Piotr Socha at P.Socha@ipczd.pl.

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
