# Peer review of "Impact of Graft Size Matching on the Early Post-Transplant Complications and Patients Survival in Children after Living Donor Liver Transplantations"

_children, 2021, doi:10.3390/children8070579_

Round 1

Reviewer 1 Report

This manuscript describes the outcomes of LDLT with a focus of graft size measured with GRWR in a single center retrospective series of more than 300 cases. Stefanowicz et al have demonstrated that good outcomes can be achieved with GRWR less than 5%, whereas adverse event can occur with GRWR >5%. The manuscript is well presented, and the topic is of interest. I have several comments which I would like to underline:

  • Materials and methods section is rather short and needs to be more specific. In particular, why did the authors divide the cohort into 4 groups and what led to the decision of GRWR thresholds (≤1.5%, >1.5% ≤3.5% etc)? i.e. why 4 groups and not two or three with different GRWR thresholds? Was this due to a statistical significance (in this case please discuss the applied methodology), arbitrary or dictated by other evidence? Please discuss this. In addition, do the authors believe that by dividing into smaller groups (for instance ≤1.5%, >1.5% ≤3.5% and >3.5) and then do a subgroup analysis they would have shown more complications and statistical significance? I am asking this as 88 patients would have had GRWR more than 3.5%, and this number is higher than some published papers. All these have to be highlighted in the methods. In the M&M, I would also add some details about immunosuppresion used after LDLT and if all rejections were biopsy proved (as the rate seems as high as 40% in the whole cohort) as well as brief statement on the management of wound closure (surgeon’s decision, anesthetics parameters, Doppler etc.).
  • Statistical analysis also needs to be more specific in the methodology used. Looking at the results, the authors stated that they have performed logistic regression model to predict the risk of mortality. This is very interesting, but the methods for this analysis have to be disclosed, i.e., the methodology used, key factors, the considered variables and what cut-off of GRWR was independently associated with risk of death etc. This is important, as some factors such as PELD, urgent LDTL etc could have contributed to the worse outcomes in GRWR >5%.
  • What is the authors position with regards to portal vein modulation (splenectomy, SA ligation etc.) in smaller grafts? Was this attempted or considered in the current study? I appreciate that this could have been a minor concern for the surgeons as form the tables provided graft sizes seem to be adequate to avoid SFSS, but a little sentence in the text on this regard would be beneficial if possible.
  • was there any software or other methods that authors used to assess estimated graft weight/GRWR before procurement? This is just my curiosity but it would be of interest for the readers.
  • Table 4: please report p values when available. These are only reported for ICU and hospital stay, whereas in the text are written for other variables (bleeding, bile leak etc). I appreciate that probably the authors reported only the ones which were statistically significant, but I believe it is worth reporting all in the table if calculated. This is also valid for the figures of Kaplan-Meier - it is worth to report the calculated p value in the figure for readers’ sake.
  • This is a retrospective, single center cohort although very interesting. The limitations of the study must be well highlighted by the authors in the discussion. Please add a paragraph in the discussion on this important point.

Minor comments:

There are some typing errors:

Abstract: LDLT should be first spelled in length and then abbreviated with LDLT. Same for HAT and PVT. If words length exceeds, I would suggest writing simply vascular complications.

Page 2, line 60: GRBW - This acronym was not addressed in the manuscript. Is this graft-recipient-body-weight or the authors meant GRWR? Same for page 9 line 278 “GRBR”.

Page 2 Line 87: the sentence “321 patients received first living donor liver transplantation and were included for further analysis” is misleading. I would simply specify that patients received primary transplant. Please shorten LDLT as this acronym is used throughout the manuscript.

Page 3 line 112. “Despite that the donor age and body mass ..” this whole sentence is repeated as it is already present in line 109 of the same page.

Page 5 line 156. “Two patients from Group B, 1 from Group C and 1 from Group D underwent liver retransplantation due to biliary complications”. The numbers are first written in words (Two) and then in number (1). The same style throughout is preferred, which is usually letters for numbers up to ten and then numbers from 10 to above.

“Table 4. Recipients posttransplant outcome following LDLT according to GRWR (n,%). Excluded are patients, who underwent retransplantation or died during hospital stay after LDLT” >>> Table 4. Recipient posttransplant outcomes following LDLT according to GRWR (n,%). Patients who underwent retransplantation or died during hospital stay after LDLT are excluded.

Page 6 line 182. Please add space between 1,5 and 10 years survival as it might seem 1.5 (one point five years). Same page line 186 yrs > years.

Page 8 line 209. These are all third person verbs (it reduces, eliminates, provides etc)

These are only some that I could spot. I believe that, overall, the manuscript would benefit from minor English revisions to improve readability.

Author Response

        Response to reviewer comments

Authors: Marek Stefanowicz, Grzegorz Kowalewski, Marek Szymczak, Waldemar Patkowski, Krzysztof Zieniewicz,  Ireneusz Grzelak, Adam Kowalski, Hor Ismail, Diana KamiĹ„ska and Piotr KaliciĹ„ski

Article Title: Impact of graft size matching on the early post-transplant complications and patients survival in children after living donor liver transplantations.

Dear reviewers:

I would like to thank you for taking the time to assess our article. We greatly appreciate the thorough and thoughtful comments provided on our submitted article. Your comments significantly improved our manuscript. It has taken us a rather long time to complete the final revision. We made sure that each one of the reviewer comments has been addressed carefully and the paper is revised accordingly.

Attached below are detailed responses to the reviewer’s comments. The latter are

shown in black and our responses in red. Please let us know if you still have any questions or concerns about the manuscript. We will be happy to address them, now in a timely manner.

Sincerely,

The authors

  • Materials and methods section is rather short and needs to be more specific. In particular, why did the authors divide the cohort into 4 groups and what led to the decision of GRWR thresholds (≤1.5%, >1.5% ≤3.5% etc)? i.e. why 4 groups and not two or three with different GRWR thresholds? Was this due to a statistical significance (in this case please discuss the applied methodology), arbitrary or dictated by other evidence? Please discuss this. In addition, do the authors believe that by dividing into smaller groups (for instance ≤1.5%, >1.5% ≤3.5% and >3.5) and then do a subgroup analysis they would have shown more complications and statistical significance? I am asking this as 88 patients would have had GRWR more than 3.5%, and this number is higher than some published papers. All these have to be highlighted in the methods. In the M&M, I would also add some details about immunosuppresion used after LDLT and if all rejections were biopsy proved (as the rate seems as high as 40% in the whole cohort) as well as brief statement on the management of wound closure (surgeon’s decision, anesthetics parameters, Doppler etc.).

We have substantially updated our materials and methods section. All the details considering statistical methods used for GRWR cut-point analysis are now described. Additional table, placed in the results section, has been created to summarize the process and visualize chosen cutoff points. To make results of our research relatable to already published data we included additional cutoff points to allow discussion on small for size grafts, even though we haven’t found sufficient evidence on impact of grafts with GRWR<1,5 on survival in our cohort.

  • Statistical analysis also needs to be more specific in the methodology used. Looking at the results, the authors stated that they have performed logistic regression model to predict the risk of mortality. This is very interesting, but the methods for this analysis have to be disclosed, i.e., the methodology used, key factors, the considered variables and what cut-off of GRWR was independently associated with risk of death etc. This is important, as some factors such as PELD, urgent LDTL etc could have contributed to the worse outcomes in GRWR >5%.

Multivariate regression analysis presentation has been significantly modified. All the details considering the initial choice of parameters and used variables in final model have been disclosed. Additional table has been created. We have also analyzed impact of GRWR on patient survival both – as a continuous variable and as a categorical variable with cut-point at 5.

  • What is the authors position with regards to portal vein modulation (splenectomy, SA ligation etc.) in smaller grafts? Was this attempted or considered in the current study? I appreciate that this could have been a minor concern for the surgeons as form the tables provided graft sizes seem to be adequate to avoid SFSS, but a little sentence in the text on this regard would be beneficial if possible.

In our study small for size syndrome is minor problem. Only three patients had GRWR < 1%.  In some recipients from Group A with GRWR ≤ 1.5% we observed hyperbilirubinemia and coagulopathy. Portal vein inflow modulation was not routinely consider in our patients. The most common indication for LDLT in this group was biliary atresia, which coexist with portal vein hypoplasia. In this patients one of major problem during LDLT is adequate portal inflow. In our institution splenectomy is procedure reserved for patients with splenomegaly and low white blood cell count ( < 2000-3000/ mm3) and platelet count ( < 50 000 /mm3). Additionally, splenectomy increases risk of PVT after LDLT.  In our cohort only in one patient  with GRWR ≤ 1.5% banding of portal vein was performed as emergency procedure due to graft congestion after reperfusion (outflow problem was excluded based on Doppler ultrasonography). 

  • was there any software or other methods that authors used to assess estimated graft weight/GRWR before procurement? This is just my curiosity but it would be of interest for the readers.

Computer tomography with three-dimensional reconstruction, ultrasound examination and magnetic resonance cholangiography were performed to evaluate hepatic vascular and biliary anatomy. In all donors biliary anatomy was also evaluated using intraoperative real-time cholangography. Currently, to estimate liver volumes we use Vitrea software (Vital® Images, Inc., U.S.), which allows for perioperative prediction of graft weight and GRWR.

  • Table 4: please report p values when available. These are only reported for ICU and hospital stay, whereas in the text are written for other variables (bleeding, bile leak etc). I appreciate that probably the authors reported only the ones which were statistically significant, but I believe it is worth reporting all in the table if calculated. This is also valid for the figures of Kaplan-Meier - it is worth to report the calculated p value in the figure for readers’ sake.

We have unified the presentation of p-values and added the missing data in Table 4 and in Kaplan Meier curves.

  • This is a retrospective, single center cohort although very interesting. The limitations of the study must be well highlighted by the authors in the discussion. Please add a paragraph in the discussion on this important point.

This study has several limitations. The main limitation is its retrospective nature. Second limitation was relatively small numbers of patients with extremely low and large GRWR in our cohort. We did not assessed recipents’ abdominal volume and differences in shape of liver grafts, especially anteroposterior thickness of graft, which is particularly problematic in smaller children without hepatomegaly or ascites. It did not allow us to assess, how these variables could affect results of using large grafts. We did not analysed in this paper late complications (hepatic artery, portal vein or biliary stenosis) and their impact on outcome.  We also were not able to determine safe lower limit of GRWR in pediatric patients.

Minor comments:

All of the suggested changes were included in the manuscript.

There are some typing errors:

Abstract: LDLT should be first spelled in length and then abbreviated with LDLT. Same for HAT and PVT. If words length exceeds, I would suggest writing simply vascular complications.

Page 2, line 60: GRBW - This acronym was not addressed in the manuscript. Is this graft-recipient-body-weight or the authors meant GRWR? Same for page 9 line 278 “GRBR”.

Page 2 Line 87: the sentence “321 patients received first living donor liver transplantation and were included for further analysis” is misleading. I would simply specify that patients received primary transplant. Please shorten LDLT as this acronym is used throughout the manuscript.

Page 3 line 112. “Despite that the donor age and body mass ..” this whole sentence is repeated as it is already present in line 109 of the same page.

Page 5 line 156. “Two patients from Group B, 1 from Group C and 1 from Group D underwent liver retransplantation due to biliary complications”. The numbers are first written in words (Two) and then in number (1). The same style throughout is preferred, which is usually letters for numbers up to ten and then numbers from 10 to above.

“Table 4. Recipients posttransplant outcome following LDLT according to GRWR (n,%). Excluded are patients, who underwent retransplantation or died during hospital stay after LDLT” >>> Table 4. Recipient posttransplant outcomes following LDLT according to GRWR (n,%). Patients who underwent retransplantation or died during hospital stay after LDLT are excluded.

Page 6 line 182. Please add space between 1,5 and 10 years survival as it might seem 1.5 (one point five years). Same page line 186 yrs > years.

Page 8 line 209. These are all third person verbs (it reduces, eliminates, provides etc)

These are only some that I could spot. I believe that, overall, the manuscript would benefit from minor English revisions to improve readability.

The manuscript underwent English revision by native speaker.

Reviewer 2 Report

Some major point I would suggest to be addressed: 

  • explanation of how the GRWR cut-offs were decided
  • I would add the p-value in all the table, even if not statistically significative
  • WIT has an important impact on the outcome after LT. It would be interesting to have data about it and any possible correlation with the outcome
  • The postoperative bleeding needing relaparotomy seems high (17%), can you compare it with the literature? do you have explanation about it? it could influence the OS and with what distribution the bleeding occured according to GRWR?
  • Can you try to explain and expande any possibile correlation with high GRWR and patient and graft survival? did you try to look at any correlation with your data? how can you say that it is not correlate with the surgical procedure?

Minor points: 

  • Before table 2, the sentences are repeated (groups, there were significant differences...)
  • Figure 3 I would add the p value

Author Response

Response to reviewer comments

Authors: Marek Stefanowicz, Grzegorz Kowalewski, Marek Szymczak, Waldemar Patkowski, Krzysztof Zieniewicz,  Ireneusz Grzelak, Adam Kowalski, Hor Ismail, Diana KamiĹ„ska and Piotr KaliciĹ„ski

Article Title: Impact of graft size matching on the early post-transplant complications and patients survival in children after living donor liver transplantations.

Dear reviewers:

I would like to thank you for taking the time to assess our article. We greatly appreciate the thorough and thoughtful comments provided on our submitted article. Your comments significantly improved our manuscript. It has taken us a rather long time to complete the final revision. We made sure that each one of the reviewer comments has been addressed carefully and the paper is revised accordingly.

Attached below are detailed responses to the reviewer’s comments. The latter are

shown in black and our responses in red. Please let us know if you still have any questions or concerns about the manuscript. We will be happy to address them, now in a timely manner.

Sincerely,

The authors

Major pionts

  • explanation of how the GRWR cut-offs were decided

We have substantially updated our materials and methods section. All the details considering statistical methods used for GRWR cut-point analysis are now described. Additional table, placed in the results section, has been created to summarize the process and visualize chosen cutoff points. To make results of our research relatable to already published data we included additional cutoff points to allow discussion on small for size grafts, even though we haven’t found sufficient evidence on impact of grafts with GRWR<1,5 on survival in our cohort.

  • I would add the p-value in all the table, even if not statistically significative

We have unified the presentation of p-values and added the missing data in Table 4 and in Kaplan Meier curves.

  • WIT has an important impact on the outcome after LT. It would be interesting to have data about it and any possible correlation with the outcome

Warm ischemia time (WIT) is used to describe two periods: 1) ischemia during organ retrieval, from the time of cross clamping left hepatic artery and left branch of portal vein  until cold perfusion is commenced, 2) ischemia during implantation, from removal of the organ from ice until reperfusion. We do not have data about ischemia when graft was procured. We have data in our registry about time of vascular anastomosis. During vascular anastomosis graft was flushed via portal vein with 2.5% cold albumin solution and kept in cold storage. It cause, that time of vascular anastomosis is different from warm ischemia time. WIT described in studies are different: Uchida WIT from 40-43 minutes [14], Kehar WIT 0.47 minutes [9] and Li WIT 1.2-1.5 minutes [8]. In study by Li et al, there was no significant difference in groups with different GRWR ( 2%<, 2-4%,>4%). It is retrospective analysis. We  do not have WIT in our database and we did not assess impact of WIT on surgical complications and patient and graft survival. We appreciate the reviewer’s insightful suggestion and agree that it would be useful to demonstrate impact of WIT on patients outcome after LDLT, but it require new analysis.

  • The postoperative bleeding needing relaparotomy seems high (17%), can you compare it with the literature? do you have explanation about it? it could influence the OS and with what distribution the bleeding occured according to GRWR?

In our patients incidence of early postoperative bleeding was 17% and was higher than described in others studies. Okada et al reported that 3.4% patients after pediatric LDLT underwent early relaparotomy  due to postoperative bleeding and in study by Hara et al 10.2% patients after adult to adult LDLT needed laparotomy due to postop-erative bleeding.  In those studies indication for relaparotomy was bleeding with hemodynamic instability. In our study additional indications for surgical intervention were: increased number of transfusion in short period of time and removal of hematoma to prevent infection complications.

  • Can you try to explain and expande any possibile correlation with high GRWR and patient and graft survival? did you try to look at any correlation with your data? how can you say that it is not correlate with the surgical procedure?

The influence of GRWR on patient and graft survival after LDLT is multifactorial with the most common cause of death in this group being multiple organ dysfunction syndrome followed by infectious complications. We try to explain this phenomenon further in the discussion part. All of the analyzed groups in our cohort (A-D) did not differ in terms of surgical complications. Moreover multivariate regression analysis model created in our research included surgical risk factors such as hepatic artery thrombosis, portal vein thrombosis, post-operative bleeding and biliary complications – nevertheless GRWR remained an independent risk factor of death in patients after LDLT.

Minor points:

All of the suggested changes were included in the manuscript.

The manuscript underwent English revision by native speaker.

Round 2

Reviewer 1 Report

Dear Authors,

Many thanks for your kind comments. All the responses to my queries have been addressed in a scientific manner and the manuscript has improved in the overall quality. The limitations of the manuscript are well-highlighted in the discussion. The authors have to be congratulated for their extensive work.

Reviewer 2 Report

Thank you for the revised version. All the main concerns have been addressed.